

# Relationship between physical and cognitive performance in community dwelling, ethnically diverse older adults: a cross-sectional study

Jennifer J. Sherwood[1], Cathy Inouye[1], Shannon L. Webb[1], Ange Zhou[2], Erik A. Anderson[1] and Nicole S. Spink[1]

[1] Department of Kinesiology, California State University, East Bay, Hayward, CA, United States of America
[2] Department of Statistics and Biostatistics, California State University, East Bay, Hayward, CA, United States of America

Corresponding author
Jennifer J. Sherwood,
jennifer.sherwood@csueastbay.edu

## ABSTRACT

**Background.** Regular exercise training stimulates physiological adaptations to improve physical performance, reduce chronic disease risk, and slow age-related cognitive decline. Since the physiological mechanisms responsible for aging-associated cognitive decline are not yet fully understood, and training-induced physiological adaptations responsible for performance measure improvements are specific to the type (aerobic vs. strength) and intensity of training, studies are needed to assess the relationships between physical performance measures and cognitive performance in older adults. These results could be used to guide exercise prescriptions with the goal of improving age-related cognitive performance. The purpose of this study was to investigate the relationship between physical performance measures and cognitive performance in a population of community dwelling, ethnically diverse older adults.

**Methods.** The cognitive performance of ninety independent, community dwelling participants (69 female, 21 male), aged 75 ± 9.5 years (mean ± SD) was measured with the Modified Mini-Mental State Test (3MS), Trailmaking Tests A and B (TMT A & B), and the Animal Naming test. Sociodemographic (age, sex, ethnicity, medication use, years of education) and anthropometric data were collected, physical activity was assessed with the Physical Activity Scale for the Elderly (PASE), peak hand-grip strength, distance walked in the 6MWT, and heart rate pre-, during, and up to 5 min. post-6MWT were measured. Forward stepwise multiple regression analyses were performed with each cognitive measure as a dependent variable.

**Results and Discussion.** Controlling for sociodemographic covariates, peak heart rate during the 6MWT (6MWT $HR_{PEAK}$) was positively correlated with performance in the 3MS ($p < 0.017$), and TMT A ($p < 0.001$) and B ($p < 0.029$). Controlling for sociodemographic covariates, PASE was positively ($p = 0.001$), and β-blocker use negatively ($p = 0.035$), correlated with performance on the Animal Naming test. Also, controlling for sociodemographic covariates, PASE was positively correlated with performance on the TMT A ($p = 0.017$). Here we show that higher peak heart rate during the 6MWT is positively correlated with cognitive performance in a population of community dwelling, ethnically diverse older adults (ages 60–95 years).

**Conclusion.** Higher peak heart rate during the 6MWT was found to be independently and positively correlated with cognitive function in community-dwelling older adults.

Although additional work is needed, these results are promising and suggest that physicians, exercise professionals, and/or fitness/fall prevention programs may use peak heart rate during the 6MWT to easily monitor exercise intensity to support cognitive health.

## INTRODUCTION

Approximately 5.4 million people, or ~22% of the population aged 71 years (yrs.) and older in the United States are affected by cognitive impairment without dementia (*Plassman et al., 2008*). Age-related chronic diseases such as hypertension and cardiovascular disease (*Ylikoski et al., 2000*), chronic atrial fibrillation (*Farina et al., 1997*), and diabetes (*Desmond et al., 1993*) are associated with impaired cognitive function. Although encouraging evidence suggests that regular physical activity reduces the risk of age-related chronic disease and slows cognitive decline (*Bherer, Erickson & Liu-Ambrose, 2013*; *Kirk-Sanchez & McGough, 2014*; *Tarumi et al., 2013*; *Barnes et al., 2003*), evidence is lacking to prescribe physical activity to benefit cognitive function beyond 'be active'. Since training-induced physiological adaptations responsible for improving physical performance measures are specific to the type (aerobic vs. strength), intensity, and duration of training, studies are needed to resolve the relationship between specific physical performance measures and cognitive performance in older adults. These results could be used to guide exercise prescriptions with the goal of improving cognitive performance. In addition, regular physical training improves health and the ease of performing daily physical activities.

In older adults, the ability to perform activities of daily living (ADL) correlates with performance in the 6MWT (*Enright et al., 2003*), a test generally well-tolerated by older adults (>68 yrs. old). The 6MWT is an exercise test in which individuals are encouraged to cover as much distance as possible at a self-selected pace, and is a measure of cardiovascular fitness and functional capacity (*Rikli & Jones, 1998*); functional capacity is the integrated responses of all the systems involved during exercise, including the pulmonary and cardiovascular systems, systemic circulation, peripheral circulation, blood, neuromuscular system, and muscle metabolism.

Positive associations between functional capacity and cognitive performance have been reported in cross-sectional studies of healthy older adults (*Matthé, Roberson & Netz, 2015*; *Lord & Menz, 2002*), but the physiological mechanisms responsible for aging-associated cognitive decline are not yet fully understood. Some studies reporting a relationship between functional capacity and cognitive performance have found that this relationship disappears after controlling for demographic covariates such as age (*Lord & Menz, 2002*) while other studies report that this relationship remains. In a cross-sectional study of 80 outpatients with stable chronic heart failure (72.4 ± 6.2) yrs.; mean ± SD), *Baldasseroni et al. (2010)* found a positive association between distance walked in the 6MWT and overall

global cognitive function assessed in the Mini Mental State Examination even after adjusting for demographic covariates, indexes of chronic heart failure severity, comorbidities, level of disability, and quality of life.

Higher hand-grip strength, an assessment of muscular fitness (*Roberts et al., 2011*), has also been associated with better age-related cognitive function (*Alfaro-Acha et al., 2006*; *Taekema et al., 2010*) and self-reported measures of physical activity (*Eggermont et al., 2009*). A longitudinal study of cognitively healthy older adults found that a higher composite measure of muscular strength was associated with a slower rate of cognitive decline (*Boyle et al., 2009*) and more specifically, maximal hand-grip strength measures have been associated with better age-related cognitive function (*Alfaro-Acha et al., 2006*; *Taekema et al., 2010*). Intervention studies report that a 16 and 9 week strength training program (*Moul, Goldman & Warren, 1995*; *Özkaya et al., 2005*) resulted in improvements to cognition, proposed mechanism being earlier sensory processing and attention to external stimuli found in the strength as compared to endurance trained group (*Özkaya et al., 2005*).

In addition to the broader dimensional aspects of exercise programs, e.g., cardiovascular vs. strength training, exercise intensity is a key component of physical training programs (*Cress et al., 2005*). Physical activity (PA) performed at a higher intensity may benefit cognitive function (*Angevaren et al., 2007*) as has been demonstrated in longitudinal (*Abbott et al., 2004*; *Laurin et al., 2001*; *Van Gelder et al., 2004*) and cross-sectional works (*Hogan, Mata & Carstensen, 2013*). In 2,257 men aged 71–93 yrs., higher intensity walking (>3 m in less than 3 s) at a baseline measure was associated with lowered risk of dementia assessed three and six years later (*Abbott et al., 2004*), and high intensity PA was associated with less cognitive loss in women (>65 yrs.) over 5 years (*Laurin et al., 2001*). Also, low-moderate intensity PA in older men followed for 10 years was found to be associated with cognitive benefits, and cognitive decline was greater in those men whose PA intensity declined over the study period (*Van Gelder et al., 2004*). Previous studies have linked distance walked during the 6MWT to cognitive function (*Matthé, Roberson & Netz, 2015*; *Lord & Menz, 2002*; *Baldasseroni et al., 2010*), but the relationship between exercise intensity during the 6MWT and cognitive performance has not been studied.

Guidelines for monitoring exercise intensity in older adults often prescribe exercise as a percentage of maximal oxygen uptake ($VO_{2max}$), percentage of heart rate reserve, Borg Rating of Perceived Exertion (RPE), or Metabolic Equivalent of Task (MET) (*Chodzko-Zajko et al., 2009*; *Nelson et al., 2007*; *Warburton, Nicol & Bredin, 2006*). For ease of use in a field test, such as the 6MWT, RPE and MET stand to be good candidates although some problems exist with application of these tests in older populations. For example, although RPE is often used as a monitoring technique, a deconditioned state and a prescribed intensity of $VO_{2max}$ may limit its accuracy (*Grange et al., 2004*; *Dunbar & Kalinski, 2004*). Additionally, direct measures of resting metabolic rates in older adults were 2.6 ml $kg^{-1}$ $min^{-1}$, 31.6% lower than the resting metabolic standard of 3.5 ml $kg^{-1}$ $min^{-1}$ normally used when calculating METs. This had the effect of elevating MET levels during walking conditions when $VO_2$ measures were being taken (*Hall et al., 2013*). Although distance covered is the primary recorded measure during the 6MWT, the additional use of

heart rate measures provides an easily obtained and objective method to monitor exercise intensity during the test.

The Physical Activity Scale for the Elderly, a self-reported index of physical activity, has also been reported to be associated with cognitive function in a large sample of 600 older participants recruited from a longitudinal cohort even when regression models were further adjusted for cardiovascular disease risk factors (*Eggermont et al., 2009*). This activity scale is a short survey that has been validated in the older population, is easily scored, and represents a composite physical activity score representing household, occupational, and recreational/exercise activities over a one-week period (*Washburn et al., 1993*).

Experimental approach to the problem:

The rationale for this study is as follows: considering the relationship between PA and cognitive performance in older adults, and the unique relationship between performance on physical measures and the type, intensity, and duration of PA, our research objective was to assess the relationships between physical and cognitive performance in community dwelling, ethnically diverse, older adults. Since sociodemographic factors are likely to moderate these relationships, multivariate analyses were used to tease out these complex relationships.

Using a cross-sectional approach, stepwise multiple regression analyses were performed to determine which physical measure best explained performance in each of the 4 dependent variables (different model for each response variable): 3MS, Trailmaking A & B, and Animal Naming test. This study suggests that peak heart rate in the 6MWT may be used to predict cognitive status.

## MATERIALS & METHODS

Participants were volunteers recruited via word-of mouth, flyers, and e-mail announcements from fitness/fall prevention programs at independent-living senior dwellings and local community senior centers, and from the general population at California State University, East Bay (CSUEB). To be included in this study, participants had to be between the ages of 60–95 yrs., nonsmokers, ambulatory (without the use of an assistive device), able to communicate (speak and read) English, and could not have uncontrolled cardiac, pulmonary, or metabolic illnesses that would contraindicate submaximal exercise testing according to guidelines established by the American College of Sports Medicine (ACSM) (*Riebe et al., 2015*). In this exploratory study, the primary objective was to determine which physical performance test best correlates with cognitive performance. Sample size was based on feasibility. Study methods and procedures were approved by the Institutional Review Board at CSUEB (#2014-231-F). Participants signed an informed consent for study participation and completed written questionnaires concerning their health, current medications, and PA level.

Although the participants studied here were asymptomatic for cardiovascular disease and otherwise physically and cognitively healthy, many reported taking prescribed medications to manage their cardiovascular health. We believe that healthy older adults are more likely to manage their health with medications so our choice to include these participants more

accurately represents most older adults. Similar to our population, a 2003 study, found that 53.2% of adults over 65 yrs. old were taking medications to manage their cardiovascular health (*Gurwitz et al., 2003*). Eligible individuals participated in a single 120-minute session during which sociodemographic and anthropometric measures, distance walked in the 6MWT, maximal hand-grip strength, cognitive performance, and heart rate pre-, during, and each minute for up to 5 min post-6MWT were measured.

## Outcome measures

Sociodemographic characteristics (age, sex, ethnicity, medication use, yrs. of education) were ascertained through self-report.

Physical activity level was determined using the Physical Activity Scale for the Elderly (PASE) which is a reliable and valid instrument for the assessment of physical activity in community dwelling older adults (*Washburn et al., 1993*).

Anthropometric measures included height, weight, and body mass index (BMI). Height was measured to the nearest 0.1 cm using a stadiometer (Seca Slider for 213 Mobile Stadiometer). Weight was measured by using a Health O Meter Professional scale (Model 500KL) and recorded to the nearest 0.1 kg. Body mass index (BMI) was calculated according to the formula: body mass (kg) divided by height squared ($m^2$).

## Cognitive measures

Cognitive performance was assessed in four cognitive tests: the 3MS, Animal Naming, and TMT A & B. To mitigate a learning effect, all participants were naïve to the cognitive tests.

The 3MS is a brief, general cognitive test battery assessing orientation, concentration, language, and immediate and delayed memory (*Teng & Chui, 1987*). Possible scores range from 0 to 102, with higher scores indicating better cognitive performance. Two questions, the first related to the suburb in which they were born and the second to the suburb in which they are located during the 3MS examination, were omitted; the maximum score was 100 points. The Animal Naming test assesses verbal fluency (*Rosen, 1980*) and participants were instructed to "name as many animals as they could think of." Performance was assessed as the total number of animal names generated in one, continuous minute.

The TMT is a measure of visual scanning, complex attention, psychomotor speed, and mental flexibility. The TMT consists of two parts A and B, each part consisting of 25 circles distributed across an 8.5 by 11-inch sheet of paper. In part A, circles are randomly arranged and numbered 1 through 25 on the page. Participants were instructed to draw a continuous line, without lifting their pencil from the paper, to connect circles in numerical sequence from 1 to 25. In part B, participants were instructed to connect alternating letters and numbers in circles arranged randomly on a page (i.e., 1-A-2-B). Tests were timed, and scores corresponded to the time to completion.

Cognitive tests were administered and scored by third and fourth year kinesiology undergraduate students who were trained on all cognitive tests by video and in-house workshops. Although no inter-tester reliability assessments were performed, the authors regularly reviewed all standardized procedures with testers and supervised data collection to ensure consistency across all tests and examiners. In addition, *Bassuk & Murphy (2003)*

reported that interrater reliability for the 3MS was high (ICC = 0.98), and there was no evidence of rater bias, when trained laypeople administered the 3MS in community populations.

## Physical performance measures

All 6MWT trials were conducted indoors according to standardized protocol (*Lipkin et al., 1986*). Prior to the 6MWT, participants were familiarized and instructed on the use of the Borg Rating of Perceived Exertion (RPE) scale. Before, and each minute during the 6-min walk, participants were shown the RPE scale and asked to "indicate their level of physical exertion" on a scale of "1 - no exertion at all" to "10- maximal exertion."

Each participant was fitted with a chest strap to monitor heart rate (Polar FS2C, Polar Electro, Oy, Kempele, Finland). Heart rate was recorded at rest after sitting quietly for 5 minutes, at each minute during the 6MWT, and each minute for 5-minutes post-exercise. The highest heart rate recorded during the 6MWT is 6MWT $HR_{PEAK}$. All participants performed one 6MWT trial. Participants were instructed to "cover as much distance as possible without jogging" during the test. During the test, participants were encouraged by having testers say, "you're doing great" and "keep up the good work." Each lapsed minute was called out to help with pacing.

Maximal hand-grip strength was assessed using a hand-grip dynamometer, recorded at 500 Hz using a Tel-100 system (Biopac Systems, Inc.) and laptop microcomputer. Specifically, participants were tested while they were standing, shoulder adducted and neutrally rotated, elbow extended, while the forearm and wrist joint were held in the neutral position, with arms against their sides (*Balogun, Akomolafe & Amusa, 1991*). Three trials were performed for each hand with a 1 minute rest between each trial. The peak value for each hand was subsequently determined via Acqknowledge version 3.5.6 software (Biopac Systems, Inc.). The maximum value using either hand was used for analyses.

## Statistical analysis

Descriptive statistics and bivariate analyses were conducted using Statistical Package for Social Sciences (version 10.0; SPSS Inc., Chicago, IL, USA). Values of $p < 0.05$ were considered to be statistically significant. Data normality was determined with Kolmogorov–Smirnov tests before analyses. Descriptive statistics presented as mean $\pm$ standard deviation (SD) for normally distributed data, median with interquartile range (IQR) for skewed data (non-Gaussian), and frequencies with percentage for categorical data. Data were skewed for age, height, BMI, 6MWT $HR_{PEAK}$, distance walked in the 6MWT, 3MS, TMT A and B, and peak hand-grip. Significant differences between male and female participants were assessed with independent Student's $t$-test for normally distributed data, while the Mann–Whitney test was used for skewed data. PASE was assessed as a continuous measure. Education levels were categorized with respect to whether the participant had more than 16 years of education (college-educated and above). Ethnicity was categorized as 1, white; 2, Asian; 3, African American; 4, other; age was measured in years, 6MWT $HR_{PEAK}$ was measured in beats per minutes (bpm) as the highest heart rate recorded during the 6MWT, and $\%HR_{max}$ was determined for each participant by calculating $HR_{max} = (220 - age)$,

then $((HR_{PEAK}/HR_{MAX}) * 100) = \%HR_{MAX}$. One participant was omitted from the 3MS analysis ($n = 89$) when a cutpoint score of 79 was applied to the 3MS scores (*Teng, Chui & Gong, 1990*). Regression diagnostics were performed to determine whether assumptions underlying the linear models were valid (linearity, equal variance, and normality of residuals), and analyses were repeated after removal of specific points with high residual values or high levels of influence. Thus, TMT A and B ($n = 88$) scores omitted for one participant, and TMT A ($n = 87$) omitted for an additional participant. Data from the 3MS was rank transformed, and the TMTs A and B were log transformed before statistical modeling to avoid violating the assumption of normality. Pearson product moment correlations (normally distributed data) and Spearman's rho correlations (skewed data) were first used to assess the relationships between cognitive performance and age, BMI, 6MWT $HR_{PEAK}$, distance walked in the 6MWT, maximal hand-grip strength, and PASE. Variables found to significantly correlate with cognitive test performance in the bivariate analyses were entered into forward multiple regression analyses with each cognitive test as a dependent variable. β-blocker use was specifically identified from medication use because it has been found to distort stress related heart rate response with the potential to affect test scores when administered to high school students (*Faigel, 1991*). Ethnicity, sex, education level and β-blocker use were suspected to contribute to cognitive performance; thus their effect was examined under a multivariate regression model setting where a class statement was used to recode the categorical variables into a set of separate binary variables, and a *contrast matrix* was constructed to assess the effect among different levels of a categorical variable. Male and female data were pooled for analyses. Model building and validation were initially compiled using Statistical Analysis System software (SAS, version 9.4) and then validated using Statistical Package for Social Sciences (Version 10.0; SPSS Inc., Chicago, IL, USA) at a .05 $\alpha$ level.

## RESULTS

Ninety-five participants ages 60–95 yrs. were recruited; however, 5 participants were excluded due to not meeting the inclusion criteria ($n = 1$), or missing test data ($n = 4$). Ninety participants (69 females) with a mean age of 75 ± 9.5 yrs. participated in this study. Nearly 58 percent of our population ($n = 52$) self-identified as white, 12 participants identified as African American, 18 identified as Asian, and 8 self-reported to be from other ethnic backgrounds. Demographic, clinical, and physical performance measures are presented in Table 1. Although males reported completing significantly more years of education (16.7 ± 3.9 vs. 14.5 ± 2.8 yrs.; $p < 0.05$) and generated significantly greater hand-grip strength (19.7 ± 7.9 vs. 11.9 ± 3.7 kg; $p < 0.001$) males vs. females, respectively), no other significant differences were found between males and females in our population. Thus, male and female data were pooled for analyses. The cognitive performance measures for all participants are shown in Table 2.

**Table 1  Participant demographic, clinical, and physical performance measures.**

|  | $n = 90$ |
|---|---|
| **Variables** | |
| Male (%) | 21 (23.33) |
| Female (%) | 69 (76.67) |
| **Continuous variables** | |
| Age (years) | 74 (67, 81.8) |
| Education (years) | $15.1 \pm 3.2$ |
| BMI (kg/m$^2$) | 25.5 (22.8, 28.8) |
| 6MWT (m) | 463.4 (387.3, 543.8) |
| 6MWT HR$_{PEAK}$ (bpm) | 112.5 (100.5, 132.3) |
| 6MWT %HR$_{MAX}$ | $74.7 \pm 11.9$ |
| RPE$_{PEAK}$ | $4.7 \pm 2.1$ |
| Peak hand grip (kg) | 13.4 (10, 15.8) |
| PASE | $162.1 \pm 78.8$ |

**Notes.**

Abbreviations:: BMI, body mass index; 6MWT, 6 Minute Walk Test; 6MWT HR$_{PEAK}$, Peak heart rate recorded during the 6MWT; 6MWT %HR$_{MAX}$, Peak heart rate during the 6MWT as a percentage of age-predicted maximal heart rate; RPE$_{PEAK}$, Peak Relative Perceived Exertion during the 6MWT; PASE, Physical activity scale of the elderly.
Data are presented as mean $\pm$ SD for normally distributed data, median (Q75, Q25) for skewed data, or n (percentage) for categorical data.

**Table 2  Cognitive test performance.**

| Continuous variables | $n$ | Range | Mean/Median |
|---|---|---|---|
| 3MS (score) | 89 | 81–100 | 96 (93.1, 95.4) |
| Animal naming (score) | 90 | 7–22 | $18 \pm 4$ |
| TMT A (s) | 87 | 15–162 | 33 (26.8, 43.8) |
| TMT B (s) | 88 | 35–280 | 80 (79, 127.3) |

**Notes.**

3MS, Modified Mini-Mental State Test; TMT, Trailmaking Test.
Data are presented as mean $\pm$ SD for normally distributed data, median (Q75, Q25) for skewed data.

## Sociodemographic, physiologic, physical performance correlates of 3MS

According to the bivariate analyses, neither BMI ($r = -0.045$, $p = 0.676$) nor PASE ($r = 0.197$, $p = 0.065$) were associated with ranked 3MS performance. However, age ($r = -0.349$, $p < 0.001$) moderately and negatively correlated with ranked 3MS performance. Distance walked in the 6MWT ($r = 0.384$, $p < 0.001$), 6MWT HR$_{PEAK}$ ($r = 0.300$, $p < 0.004$), and peak hand-grip strength ($r = 0.297$, $p = 0.006$) were moderately and positively correlated with ranked 3MS performance, and thus were entered into the final model. Table 3 shows the final model which explained 33.3% of the total variance in 3MS scores ($R^2 = 0.333$, $F(3, 85) = 15.677$, $p < 0.001$). After adjusting for sociodemographics (age, ethnicity, education) and β-blocker use, higher 6MWT HR$_{PEAK}$ was associated with higher 3MS scores.

**Table 3 Stepwise multiple regression model with 3MS as the dependent variable.**

| Variable | Unstandardized coefficient (B) | SE | 95% CI | Standardized coefficient (β) | p-Value |
|---|---|---|---|---|---|
| Intercept | 52.338 | 29.583 | | | 0.080 |
| White | 23.902 | 4.559 | 14.837 to 32.967 | 0.460 | <0.001 |
| Age | −0.760 | 0.264 | −1.285 to −0.234 | −0.279 | 0.005 |
| 6MWT HR$_{PEAK}$ | 0.306 | 0.125 | 0.056 to 0.555 | 0.238 | 0.017 |

Notes.
Modified Mini-Mental State Test (3MS); 3MS is rank transformed prior to model fitting; SE, Standard Error; 95% CI, 95% Confidence Intervals for the coefficients; Peak heart rate recorded during the 6-Minute Walk Test (6MWT HR$_{peak}$).

**Table 4 Stepwise multiple regression model with animal naming as the dependent variable.**

| Variable | Unstandardized coefficient (B) | SE | 95% CI | Standardized coefficient (β) | p-Value |
|---|---|---|---|---|---|
| Intercept | 14.400 | 0.940 | | | <0.001 |
| Asian | −3.044 | 0.902 | −4.837 to −1.250 | −0.304 | 0.001 |
| PASE | 0.015 | 0.005 | 0.006 to 0.024 | 0.299 | 0.001 |
| Education | 2.082 | 0.746 | 0.599 to 3.565 | 0.260 | 0.006 |
| β-blocker | −2.069 | 0.964 | −3.986 to −0.152 | −0.198 | 0.035 |

Notes.
SE, Standard Error; 95% CI, 95% Confidence Intervals for the coefficients; Physical Activity Scale for the Elderly (PASE).

## Sociodemographic, physiologic, physical performance correlates of Animal Naming

Correlations revealed that neither BMI ($r = 0.125$, $p = 0.242$) nor peak hand-grip ($r = 0.125$, $p = 0.253$) correlated with Animal Naming performance, but age ($r = −0.355$, $p < 0.001$) was moderately and negatively correlated with Animal Naming test performance. Distance walked in the 6MWT ($r = 0.395$, $p < 0.001$), PASE ($r = 0.390$, $p < 0.001$), and 6MWT HR$_{PEAK}$ ($r = 0.242$, $p = 0.022$) were moderately and positively correlated with Animal Naming test performance and thus, were entered into the final model. Table 4 shows the final model which explained 32.1% of the total variance ($R^2 = 0.321$, ($F(4, 85) = 11.496$, $p < 0.001$). After adjusting for sociodemographics (age, ethnicity, education) and β-blocker use, PASE independently and positively correlated with Animal Naming performance ($p$-value $= 0.001$). Higher levels of self-reported PA on the PASE were associated with better performance on the Animal Naming test. In contrast, β-blocker use was found to negatively correlate with Animal Naming performance ($p$-value $= 0.035$). On average, self-reported β-blocker users scored 2.1 points lower on the Animal Naming test, compared with non-users.

## Sociodemographic, physiologic, physical performance correlates of TMT A

Correlations revealed that neither BMI ($r = 0.070$, $p = 0.522$) nor peak hand-grip ($r = −0.001$, $p = 0.996$) correlated with log-transformed Trailmaking A performance. However, age ($r = 0.411$, $p < 0.001$) was positively, while education level ($r = −0.356$, $p = 0.001$), PASE ($r = −0.341$, $p = 0.001$), 6MWT HR$_{PEAK}$ ($r = −0.407$, $p < 0.001$), and

**Table 5  Stepwise multiple regression model with TMT A as the dependent variable.**

| Variable | Unstandardized coefficient (B) | SE | 95% CI | Standardized coefficient ($\beta$) | p-Value |
|---|---|---|---|---|---|
| Intercept | 3.608 | 0.543 | | | <0.001 |
| 6MWT HR$_{peak}$ | −0.005 | 0.002 | −0.012 to −0.003 | −0.214 | <0.001 |
| PASE | −0.005 | 0.001 | −0.002 to 0.00 | −0.226 | 0.017 |
| Age | 0.011 | 0.005 | 0.001 to 0.020 | 0.234 | 0.031 |

**Notes.**

TMTA,  Trailmaking Test A; TMT A is lognormal distributed.
A log transformation was taken prior to model fitting; SE, Standard Error; 95% CI, 95% Confidence Intervals for the coefficients; Peak heart rate recorded during the 6-Minute Walk Test (6MWT HR$_{peak}$).

**Table 6  Stepwise multiple regression model with TMT B as the dependent variable.**

| Variable | Unstandardized coefficient (B) | SE | 95% CI | Standardized coefficient ($\beta$) | p-Value |
|---|---|---|---|---|---|
| Intercept | 4.426 | 0.605 | | | <0.001 |
| 6MWT HR$_{peak}$ | −0.006 | 0.003 | −0.011 to −0.001 | −0.236 | 0.029 |
| Education | −0.250 | 0.098 | −0.444 to −0.056 | −0.254 | 0.012 |
| Age | 0.011 | 0.006 | 0.000 to 0.022 | 0.210 | 0.049 |

**Notes.**

Trailmaking Test B (TMT B); TMT B is lognormal distributed. A log transformation was taken prior to model fitting; SE, Standard Error; 95% CI, 95% Confidence Intervals for the coefficients; Peak heart rate recorded during the 6-Minute Walk Test (6MWT HR$_{peak}$).

distance walked on the 6MWT ($r = -0.418$, $p < 0.001$) were moderately and negatively correlated with TMT A time to completion. Table 5 shows the final model which explained 25.2% of the total variance ($R^2 = 0.252$, ($F(3, 83) = 10.644$, $p < 0.001$). Controlling for sociodemographics (age, ethnicity, education), both PASE scores and 6MWT HR$_{PEAK}$ negatively correlated with log-transformed TMT A performance with $p$-values $= 0.017$ and <0.001, respectively. This means that higher 6MWT HR$_{PEAK}$ and PASE scores were associated with a faster TMT A time to completion, or better test performance.

## Sociodemographic, physiologic, physical performance correlates of TMT B

Correlations revealed that neither BMI ($r = 0.070$, $p = 0.522$), PASE ($r = -0.142$, $p = 0.186$), nor peak hand-grip ($r = -0.017$, $p = 0.877$) correlated with log-transformed Trailmaking B performance. However, age ($r = 0.387$, $p < 0.001$) was moderately and positively associated with TMT B time to completion while education level ($r = -0.406$, $p < 0.001$), 6MWT HR$_{PEAK}$ ($r = -0.411$, $p < 0.001$), and 6MWT distance ($r = -0.328$, $p = 0.002$) were moderately and negatively associated with TMT B time to completion. Thus, these variables were entered into the final model (Table 6) and accounted for 25.1% of the total variance ($R^2 = 0.251$, ($F(3, 84) = 10.739$, $p < 0.001$). Controlling for sociodemographics (age, ethnicity, education level), 6MWT HR$_{PEAK}$ was negatively correlated with TMT B time to completion with a $p$-value <0.05. This means that higher 6MWT HR$_{PEAK}$ was associated with a faster TMT B time to completion, or better test performance.

## DISCUSSION

Main findings of the current study were: (1) exercise intensity (6MWT $HR_{PEAK}$) recorded during the 6MWT was correlated with better performance on a range of cognitive tests (3MS and TMT A & B); this correlation remained after adjusting for sociodemographics, (2) higher PA levels reported on the PASE was correlated with better performance on verbal fluency (Animal Naming) and TMT A, (3) distance walked in the 6MWT was associated with better performance on cognitive tests; however, these relationships were no longer evident after controlling for sociodemographics (age, ethnicity, education), (4) no relationship was found between maximal hand-grip strength and performance on cognitive tests, and (5) a negative correlation was found with β-blocker use and verbal fluency (Animal Naming).

In the current study, higher $HR_{PEAK}$ during the 6MWT was positively correlated with cognitive performance in the 3MS and TMT A and B. These findings are in agreement with evidence that has shown intensity of PA positively influences cognitive abilities (*Van Gelder et al., 2004*; *Angevaren et al., 2007*; *Brown et al., 2012*). Longitudinal studies found that higher self-reported PA intensity preserved cognitive function in middle aged and older adults tested over a four (*Angevaren et al., 2007*) and ten-year follow-up period (*Van Gelder et al., 2004*). In a cross-sectional study, direct measures of daily PA intensity were assessed with an Actigraph monitor and those in the higher intensity tertiles scored higher on cognitive tests (*Brown et al., 2012*). Our study findings are unique in that peak heart rate recorded during the 6MWT provided a direct and easily obtained measure of exercise intensity that was correlated with cognitive performance measures.

In the bivariate correlations performed here, distance walked in the 6MWT was correlated with cognitive performance (3MS, Animal Naming, and TMT A & B), but after adjusting for confounding variables such as age, distance walked was no longer associated with global cognitive functioning. These results are consistent with those of *Lord & Menz (2002)* in which the association between cognitive and physical performance disappeared after controlling for age. *Matthé, Roberson & Netz (2015)* included participants using walking aids, and found that those participants who did not use walking aids, covered more distance in the 6MWT and scored higher on the MMSE. However, sample size was much smaller ($n = 27$) and confounding variables such as age were not controlled. In addition, our population may have been more physically homogeneous than populations in previous studies, thus associations with PA may have been more difficult to resolve in this study.

Physical activity as self-reported in the PASE maintained a positive association with Animal Naming performance after controlling for age, ethnicity, and education level. This association might be partially explained by the positive influence of PA on psychomotor function in older adults (*Rodríguez-Aranda et al., 2006*; *Rosano et al., 2010*). Evidence of an age-associated slowing of psychomotor function has been demonstrated as the slowing of handwriting, reading speed and time required to say a particular word in older participants (*Eggermont et al., 2009*). Additionally, PA may be associated with the frequency of social interactions (e.g., group exercise, exercise with a friend) and a higher PASE score may

reflect less social isolation. This is in part supported by *McAuley et al. (2003)* who found self-reported PA levels assessed by PASE to be influenced by perceptions of social support with higher perceived support associated with increased self-efficacy and improved exercise adherence.

Maximal hand-grip strength was not found to be associated with cognitive performance, a finding that remains equivocal in the literature. Systematic reviews support the use of hand-grip measures to predict numerous outcomes, including mortality, functional limitations, activity levels, and independence, these measures most likely reflecting multidimensional variables that contribute to future risk in older adults (*Bohannon, 2001*; *Bohannon, 2008*). Cross-sectional and longitudinal studies suggest that muscular strength measures are positively associated with cognitive performance (*Enright et al., 2003*; *Lord & Menz, 2002*; *Alfaro-Acha et al., 2006*; *Taekema et al., 2010*; *Ling et al., 2010*). Longitudinal studies point to hand-grip as a significant predictor of cognitive decline in 555 participants, 85 years of age, followed for 4 years (*Taekema et al., 2010*) and in 2,160 older Mexican Americans 65 years and older followed for a 7-year period (*Alfaro-Acha et al., 2006*), however this association was no longer significant after controlling for age (*Lord & Menz, 2002*). Also, some participants in our study complained of limited ability to perform the hand-grip strength test due to arthritic conditions, which may have limited our ability to detect an association between grip strength and cognitive performance. Perhaps with a larger sample and/or a longitudinal study, a temporal association would emerge.

The negative correlation between β-blocker use and verbal fluency is an interesting one and warrants further investigation. Similar to results from *Qato et al. (2008)*, ∼83% (75 of 90 participants) reported regular use of medications to manage chronic health conditions, and ∼21% of these participants (16 of 75 participants) reported using β- blockers. Beta blockade has been found to abolish the stress induced rise in heart rate (*Benschop et al., 1994*) and when administered to students with paralyzing test anxiety, improved Scholastic Aptitude Test scores (*Faigel, 1991*). Further benefit has been attributed to tests of cognitive flexibility (*Alexander et al., 2007*), but the authors are not aware of research links to verbal fluency.

Although the current study was not designed to test a specific mechanism, possible explanations exist and justify further research. Regular exercise at higher intensities improves cardiovascular health, cardiorespiratory fitness, lipid profiles, and is believed to benefit cerebral blood flow (*Van Gelder et al., 2004*; *Kraus et al., 2002*; *Rogers, Meyer & Mortel, 1990*), a theory in part supported by a cross-sectional study showing that cerebral blood flow reductions in sedentary older adults are associated with impaired cognitive performance (*Rogers, Meyer & Mortel, 1990*). Additional mechanisms include the exercise-stimulated rise in growth factors believed to promote hippocampal synaptic plasticity and neurogenesis (*Cotman, Berchtold & Christie, 2007*; *Gómez-Pinilla, So & Kesslak, 1998*; *Van Praag et al., 1999*).

In our study, a higher peak exercise intensity during the 6MWT was associated with better cognitive performance. The 6MWT is a test of functional capacity, and as such depends on the health and functioning of multiple systems (*Stewart et al., 2000*) including the cardiovascular and muscular systems. Our results show that a higher exercise

intensity during an acute exercise test (i.e., the 6MWT) is correlated with better cognitive performance. We assume that higher heart rate during the 6MWT is indicative of chronic physical training effects on the heart. However, additional studies are needed to test whether chronically training at higher intensities benefits cognitive performance in older adults.

In this population of ethnically diverse older adults, ethnicity correlated with performance in the 3MS and Animal Naming even after controlling for self-reported education level. Work by *Manly et al. (2003)* suggests that in multi-ethnic older adults, cognitive test performance should be interpreted relative to education quality, not years of formal education. In a population of ethnically diverse community dwelling older adults, *Manly et al. (2002)* found that African American older adults obtained significantly lower scores than whites on measures of word list learning and memory, figure memory, abstract reasoning, fluency, and visuospatial skill even though the groups were matched on years of education. However, after adjusting for education quality, the overall effect of ethnicity was greatly reduced and ethnicity differences on tests became nonsignificant. Culture, ethnicity, and poverty affect education quality so that years of education and quality of education are not comparable in a population of multi-ethnic older adults (*Manly et al., 2005*).

It is worth noting that despite reaching significance, the standardized coefficients (β-weights) indicated a moderate ability to predict cognition, a finding that is consistent with other cross-sectional (*Sprague, Watts & Burns, 2016*; *Hultsch, Hammer & Small, 1993*; *Baldasseroni et al., 2010*; *Angevaren et al., 2007*) and longitudinal studies (*Aichberger et al., 2010*). Distance walked during the 6MWT was found to be a significant predictor of MMSE in patients with chronic heart failure (β = 0.20) (*Baldasseroni et al., 2010*), and selected PASE items predicted cognition (β-weights ranging from 0.30–0.31) (*Sprague, Watts & Burns, 2016*) as well as self-reported measures of activity frequency (β-weight of 0.019–0.262 (*Hultsch, Hammer & Small, 1993*). *Angevaren et al. (2007)* and *Aichberger et al. (2010)* found intensity to be predictive of cognition with β-weights that ranged from 0.16–0.46. These findings are consistent with ours and suggest that the relationship between physical activity or exercise intensity, and cognitive performance is complex and not fully understood, with ethnicity in the current study modifying this relationship. In conclusion, although the 6MWT is a practical measure, there are inherent limits as to what can be assessed and additional studies are needed to elucidate other contributors.

Study limitations include the cross-sectional study design and homogeneity of our population. It is possible that physical exercise is a marker for a healthy lifestyle; better overall health linked to cognitive health, however our data are unable to ascribe direction of causality. Additionally, many of our participants were recruited from local fall prevention/fitness programs and thus likely to represent a segment of more active persons aged 60 to 95 years. Other limitations include that we did not test for educational quality in our sample of multi-ethnic older adults, and facility and participant scheduling constraints precluded us from offering an acclimation period for the 6MWT. Finally, while our findings show that a higher intensity (HR $_{PEAK}$) during the 6MWT was associated with better performance on a range of cognitive tests, we did not control the intensity at which participants walked but allowed them to self-pace during the 6MWT.

## CONCLUSION

Present study found a lack of association between grip strength and distance walked, and cognition, and β-blocker use negatively correlated to verbal fluency. Authors speculate that arthritis in participants' hands limited grip strength, and our homogeneous sample prevented distance walked from being isolated as a contributor to cognitive performance. Additionally, authors are not aware of previous research linking β-blocker use to verbal fluency, although *Faigel (1991)* reports a reduction in test anxiety and improved Scholastic Aptitude Scores. These findings warrant future testing that controls for these variables. Self-reported PASE measures were associated with Animal Naming perhaps reflecting reduced social isolation and slowing of psychomotor function associated with physical activity.

   Key to this study, was to identify safe, inexpensive and field-amenable methods to monitor cognition in community dwelling older adults. A submaximal field test, such as the 6MWT, may be part of that solution. Peak exercise intensity during the 6MWT is a simple and reliable measure of functional capacity, and is related to cognitive performance in multi-ethnic, community dwelling older adults (ages 60–95 yrs.). This is promising because it suggests that a submaximal field test may be used to monitor age-related changes in executive function and higher order processing. Thus, physicians, exercise professionals, and/or fitness/fall prevention programs could use this short, physical test to monitor cognition and motivate people to remain active with age.

## ACKNOWLEDGEMENTS

Thank you to: Elizabeth Johnson, Brendan Jordan, Kevin Medina, Joaquin Tabera, Nang Ei Ei Mon, Ritika Vashisht, Cindayanne Camarse, Sherri Morioka, Roger Chandler, and Christopher Bayesdorfer, PhD.

### Funding

This work was made possible with California State University, East Bay (CSUEB) Faculty Support Grants to Sherwood & Inouye and CSUEB Center for Student Research Fellowships to Erik A. Anderson and Nicole S. Spink. The funders had no role in study design, data collection and analysis, decision to publish, or preparation of the manuscript.

### Grant Disclosures

The following grant information was disclosed by the authors:
California State University.
East Bay (CSUEB) Faculty Support.
Sherwood & Inouye.
CSUEB Center for Student Research Fellowships.

### Competing Interests

The authors declare there are no competing interests.

## Author Contributions

- Jennifer J. Sherwood, Cathy Inouye and Shannon L. Webb conceived and designed the experiments, performed the experiments, analyzed the data, contributed reagents/materials/analysis tools, prepared figures and/or tables, authored or reviewed drafts of the paper, approved the final draft.
- Ange Zhou analyzed the data, contributed reagents/materials/analysis tools, prepared figures and/or tables, authored or reviewed drafts of the paper, approved the final draft.
- Erik A. Anderson and Nicole S. Spink conceived and designed the experiments, performed the experiments, authored or reviewed drafts of the paper.

## Human Ethics

The following information was supplied relating to ethical approvals (i.e., approving body and any reference numbers):

The Institutional Review Board at California State University, East Bay granted ethical approval to carry out the study within its facilities (Ref: 2014-231-F).

## Data Availability

Raw data is available as Data S1.

## Supplemental Information

Supplemental information for this article can be found online at http://dx.doi.org/10.7717/peerj.6159#supplemental-information.

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
