# Peer review of "Relationship between physical and cognitive performance in community dwelling, ethnically diverse older adults: a cross-sectional study"

_PeerJ, doi:10.7717/peerj.6159_

## Round 0.1 · original submission · Major Revisions

I would like to invite the authors to revise the manuscript according to the reviewers comments.

Reviewer 1 ·

Basic reporting

1. The authors used a clear English language.
2. Raw data was supplied.
3. Tables have a good quality and are relevant.
4. The background presented in the introduction section needs improvement. A contextualization should be done to justify the use of the covariates in the analyses performed. Additionally, the gap existent in the literature considering the relation between hand-grip strength and cognitive function needs to be presented.

Experimental design

1. The research question is well defined. However, two points need to be improved.
- The existent gap in the literature that is filled by the study needs to be clarified.
- Some analyses presented in the results section were not the aim of the study. The aim presented was: “to examine the relationship between physical performance measures, such as the exercise intensity and distance walked in the 6MWT, hand-grip strength, and cognitive performance parameters in community dwelling, ethnically diverse, older adults”. However, the relationship between cognitive performance were also investigated with other parameters (e.g. PASE, age, B-blocker use, etc.). Please, clarify why these analyses were done.
2. Some aspects should be improved in the Methods section:
- Before using the acronym RPE, it is necessary to show what it means. It is also important to inform whether subjects were familiarized to the RPE scale use.
- Please describe the instructions given to the participants regarding the 6MWT.
- Considering the hand-grip strength, it is not clear which value was used in the analyses (was it a mean of the peak values for each hand? was it the peak value of both hands, independent of the hand that presented the peak value?).
- Please, explain how is the score of the 3MS test given.
- For the categorical parameters, which statistical test was used to investigate the associations with the cognitive performance?

Validity of the findings

1. Results for all correlations performed should be presented in the paper. Only the significant correlations were presented in the Results section. Please, present all values.
2. Despite the significant correlations found for some parameters, all of them were only weak or moderate. This fact should be considered in the Discussion section and taken into account in the Discussion and Conclusion.
3. The Conclusion is based mainly in the results of the relationship between cognitive performance and peak exercise intensity during the 6MWT. All physical performance measures should be considered, to answer the research question presented in the aim of the study.

Additional comments

1. The background presented in lines 1-4 in the Abstract does not contextualize the aim of the study. Please, revise it.
2. Please, change +/- to ± in the Abstract.
3. As commented previously, some results presented in the Abstract were not the aim of the study. Please, clarify why these analyses were done.
4. Review the p value presented in the Abstract for the relation between the peak heart rate during the 6MWT and the 3MS test (p<0.071).
5. Please, clarify the sentence in lines 102 to 105 in the introduction section. It is confuse due to the use of the terms “baseline”, and “three and six years later”.
6. The parameters 6MWT %HRMAX should be described in the Methods section, before using it in the Table 1.
7. In the Table 2, the n for each cognitive test is different (89-90-87-88). Why are they different? In the Results section the authors described that from 95 participants, 5 were excluded due to not meeting the inclusion criteria or missing data. Accordingly, data from 90 participants should be presented.

Reviewer 2 ·

Basic reporting

The article is simple with the use of simple tools but that brings data applicable in practice. One suggestion is to make a theoretical foundation involving more the heart rate as a control of the intensity of the training with the elderly and to bridge the cognitive that is already well documented that must be moderated for both acute and chronic effects, because the works that the authors cite in the introduction they evaluate intensity with other variables.

Experimental design

The inclusion and exclusion criteria should be clearer as this population receives a fall prevention intervention.
The authors can not say that they controlled the intensity of the exercise, but that they controlled the intention of the FC during exercise. To control the intensity of the exercise we have to control other variables.

Validity of the findings

Data is replicable

---

## Round 0.2 · accepted · Accept

Dear authors. Congratulations on your accepted paper.

# Reviewer 1 ·

Basic reporting

The authors addressed all aspects pointed out in the review.

Experimental design

The authors addressed all aspects pointed out in the review.

Validity of the findings

The authors addressed all aspects pointed out in the review.

Additional comments

The authors addressed all aspects pointed out in the review.